# Paternal Exposure to Bisphenol-A Transgenerationally Impairs Testis Morphology, Germ Cell Associations, and Stemness Properties of Mouse Spermatogonial Stem Cells

**DOI:** 10.3390/ijms21155408

**Published:** 2020-07-29

**Authors:** Polash Chandra Karmakar, Jin Seop Ahn, Yong-Hee Kim, Sang-Eun Jung, Bang-Jin Kim, Hee-Seok Lee, Sun-Uk Kim, Md Saidur Rahman, Myung-Geol Pang, Buom-Yong Ryu

**Affiliations:** 1Department of Animal Science and Technology and BET Research Institute, Chung-Ang University, Anseong 17546, Korea; polashmicro@gmail.com (P.C.K.); ahnjs@cau.ac.kr (J.S.A.); yhkcau@naver.com (Y.-H.K.); tkddms2428@naver.com (S.-E.J.); shohagvet@gmail.com (M.S.R.); mgpang@cau.ac.kr (M.-G.P.); 2Department of Cancer Biology, Perelman School of Medicine, University of Pennsylvania, Philadelphia, PA 19104, USA; bakim430@gmail.com; 3Department of Food Science & Technology, Chung-Ang University, Anseong 17546, Korea; hslee0515@cau.ac.kr; 4National Primate Research Center and Futuristic Animal Resource & Research Center, Korea Research Institute of Bioscience and Biotechnology (KRIBB), Ochang 28116, Korea; sunuk@kribb.re.kr

**Keywords:** bisphenol A, testis morphology, germ cell associations, apoptosis

## Abstract

Bisphenol-A (BPA) exposure in an adult male can affect the reproductive system, which may also adversely affect the next generation. However, there is a lack of comprehensive data on the BPA-induced disruption of the association and functional characteristics of the testicular germ cells, which the present study sought to investigate. Adult male mice were administered BPA doses by gavage for six consecutive weeks and allowed to breed, producing generations F1–F4. Testis samples from each generation were evaluated for several parameters, including abnormal structure, alterations in germ cell proportions, apoptosis, and loss of functional properties of spermatogonial stem cells (SSCs). We observed that at the lowest-observed-adverse-effect level (LOAEL) dose, the testicular abnormalities and alterations in seminiferous epithelium staging persisted in F0–F2 generations, although a reduced total spermatogonia count was found only in F0. However, abnormalities in the proportions of germ cells were observed until F2. Exposure of the male mice (F0) to BPA alters the morphology of the testis along with the association of germ cells and stemness properties of SSCs, with the effects persisting up to F2. Therefore, we conclude that BPA induces physiological and functional disruption in male germ cells, which may lead to reproductive health issues in the next generation.

## 1. Introduction

Bisphenol-A (2,2-bis(4-hydroxyphenyl)propane; BPA) is one of the most commonly used endocrine disruptors worldwide, with widespread industrial applications in the manufacture of plastics and epoxy resins [1,2]. It is chemically stable and has been shown to exhibit environmental persistence after leaching from BPA-enriched products at normal or elevated temperature [3,4]. BPA may show bioaccumulation in animals and has been detected in human saliva, blood, plasma, amniotic fluid, placental tissue, breast milk, urine, follicular fluid, and adipose tissue [5,6]. High urinary BPA levels are associated with a reduction in the number of follicles and ovarian volume in women [7]. In males, urinary BPA may be linked to decreased semen quality and increased DNA damage in spermatozoa [8]. Furthermore, other studies have demonstrated the anti-androgenic and estrogenic properties of BPA, both in vitro and in vivo [9,10]. Therefore, BPA can exert widespread effects on several endocrine-related biological pathways and reproductive organs [11,12] and exhibit cellular toxicity toward many cell types, including male germ cells [13,14,15,16].

Spermatozoa are produced from spermatogonial stem cells (SSCs) as the final products of spermatogenesis, which appear that the balance of self-renewal and differentiation should be directly related to the SSCs [17,18]. BPA possibly disrupts endocrine signaling owing to its estrogenic activity [19] and interferes with spermatogenesis. Therefore, several studies have been conducted on the correlation between exposure to BPA and poor sperm characteristics. These results show that including low quality [20], increased DNA damage [21], and alterations in its fertility-related protein levels through intracellular protein profiles [22], of spermatozoa in vitro [23]. Additional studies have attempted to investigate the BPA-induced disruption of meiotic progression [24] and chromosome dynamics [25], aberrant DNA methylation [26], and oxidative stress [27] during spermatogenesis. The adverse health effects associated with BPA have led to the formulation of guidelines regarding permissible exposure limits. As per the guidelines issued by the US Environmental Protection Agency (EPA), the proposed values of the no-observed-adverse-effect-level (NOAEL) and the lowest-observed-adverse-effect level (LOAEL) for BPA exposure are 5 mg/kg body weight (bw)/day and 50 mg/kg bw/day, respectively [28].

Previous studies have shown that BPA is involved in the alteration of the proteome and functional properties of SSCs and testicular germ cells, along with Sertoli cells [29,30]. Further, in vivo studies on the effect of gestational exposure of mice to EPA-proposed BPA doses have shown that it leads to poor sperm characteristics and alterations in the proteome profile in adults of the F1 generation [31]. Low dose neonatal BPA exposure of neonate male mice was shown to cause meiotic arrest and apoptosis of germ cells [32]. Exposure to BPA was also found to have adverse effects on the mitochondria of testicular cells, resulting in apoptosis and compromised reproductive function in adult rodents [33,34]. Moreover, exposure of male rodents to BPA resulted in a reduction of the size of the epididymides, along with a decreased efficiency of sperm production [35,36]. Thus, the wide-ranging effects of BPA on reproductive health have been well-studied, especially in relation to the genetics and proteomics of germline cells, tissues, and organs. Hence, it is necessary to assess the transgenerational effects of BPA exposure in adult age on testis morphology, association of germ cells in testes, and functional properties of SSCs. In the current study, two reference doses of BPA (NOAEL and LOAEL) were applied to adult male mice (F0) for six weeks, and a breeding scheme with outbred females was applied to produce generations (F1–F4). The entire experiment design is illustrated in Figure 1A,B. The offspring of all generations were checked for abnormalities in the morphology of the testis, germ cell staging pattern in seminiferous epithelium (SE), and proportion of germ cells in SE. BPA-induced apoptosis in testicular cells was also evaluated across generations. Finally, stemness properties of BPA-exposed SSCs were measured by transplanting germ cells into the testis of recipient mice.

## 2. Results

### 2.1. BPA Increases the Frequency of Abnormal Seminiferous Tubules (STs) in F0–F2 Generations

For testicular abnormalities, H&E stained testis sections of F0–F4 generations were examined. Abnormal seminiferous tubules (STs) were characterized as abnormal in the case of huge lumen size, abnormal cell mass in the lumen, and presence of abnormal vacuoles resulting in germ cell loss or STs that lack lumen (Figure 2A). We observed a significantly high frequency of abnormal STs in the BPA-exposed groups (F0) (Figure 2B) at both NOAEL and LOAEL doses. This phenomenon persisted, and a relatively high percentage of abnormality was found in F1 generation at the LOAEL dose and in F2 generation at both the NOAEL and LOAEL doses (Figure 2C,D). However, we did not observe any significant changes in the percentage of abnormal STs in F3 and F4 generations (Figure 2E,F).

### 2.2. BPA Changes the Size of SE and Alters Stages of STs

A high frequency of BPA-induced abnormal STs was observed in F0, which persisted transgenerationally in F1 and F2. These findings led us to investigate the effect of BPA on the size of SE and the spermatogenesis stages inside the STs. We measured the area of STs and lumens of all generations. Although no remarkable changes were seen in the ST area in F0 (see Appendix A) and F1 (see Appendix A), a significantly greater lumen area was observed in stages VII and VIII both in F0 (see Appendix A) and in F1 (see Appendix A) in the LOAEL-exposed groups. A higher lumen area, in fact, represented the thinner SE area because the sizes of STs in all groups were similar. Additionally, we examined the area of the lumen in F2 (see Appendix A), F3 (see Appendix A), and F4 (see Appendix A) but did not observe any significant changes. Next, we scrutinized the SE stages and found that the percentage of stage VII increased significantly in F0, F1, and F2 generations, especially at the LOAEL dose (Figure 3A–C), but that of stage VIII decreased dramatically in these generations exposed to the same dose (Figure 3A–C). However, no significant differences in the SE stages were observed in F3 (Figure 3D) and F4 (Figure 3E) generations. Additionally, significant differences in SE stages were also observed in stage I, III, IV, and VI in F1 generation due to BPA exposure (Figure 3B).

### 2.3. BPA-Induced Alterations in the Count of Spermatogonia

The observation that BPA exposure caused a decrease in the area of the SE in F0 and F1 led us to investigate the changes in the total count of spermatogonia in comparison to the number of Sertoli cells (Figure 4A). We observed a significant reduction in spermatogonia/Sertoli cell ratio at the LOAEL in the BPA-exposed group (F0) (Figure 4B). We did not observe any difference in this ratio in the subsequent generations (F1–F4) (Figure 4C–F).

Next, we performed FACS assay to evaluate effect of BPA on the total number of testicular germ cell subpopulations. The results showed three peaks, depending on the DNA content of the cells (1C, 2C, and 4C cells). The type 1C cells represent spermatids, while type 2C cells indicate somatic cells, spermatogonia, and secondary spermatocytes. The type 4C cells represent cells in the G2/M phase including primary spermatocytes (Figure 5A). It was observed that the LOAEL group of F0 generation showed a significant increase in the total spermatid number (Figure 5B) and a remarkable reduction in the number of 2C-type cells (Figure 5C) and 4C-type cells (Figure 5D). A similar scenario was observed in F1; 1C-type cells increased at the LOAEL (Figure 5E), while 2C-type cells decreased both at the NOAEL and LOAEL (Figure 5F), and 4C-type cells also decreased at the LOAEL (Figure 5G). The 1C-type cells increased only at the LOAEL in F2 (Figure 5H), and no effects were observed on the other cell types in this generation (Figure 5I,J). The population proportion of testicular germ cells was also studied in the subsequent generations, but BPA-induced effects were not found in F3 and F4 (see Appendix A).

### 2.4. BPA Exposure Induces Germ Cell Apoptosis

The increase in the number of apoptotic germ cells on BPA exposure was studied using the TUNEL assay (Figure 6A). The results showed a significant increase in the percentage of apoptotic tubules and TUNEL-positive germ cells in both NOAEL- and LOAEL-exposed groups of F0 generation (Figure 6B,G). Additionally, the percentage of apoptotic tubules and apoptotic germ cells per tubule increased in F1 at the LOAEL (Figure 6C,H). In the F2 generation, a high frequency of apoptotic tubules was found at both the NOAEL and LOAEL (Figure 6D), and a high number of apoptotic cells were found in the LOAEL-exposed group (Figure 6I). However, the subsequent generations recovered from this trend of germ cell loss, and no significant changes were observed in the percentage of apoptotic tubules and the number of apoptotic cells per tubule in F3 (Figure 6E,J) and F4 (Figure 6F,K) generations.

### 2.5. BPA Affects the Stemness Properties of SSCs in F2 Offspring

We observed that BPA-induced effects persisted mostly up to F2 generation and diminished in the subsequent generations. Therefore, we decided to transplant the germ cells from F2 and F3 into the testes of recipient CD1 mice, to observe BPA-related effects on the stemness properties of SSCs. One month after transplantation, germ cells labeled with PKH26 were visualized under a fluorescence microscope (Figure 7A), and donor-derived colonies were counted. We observed a significantly low colony count in F2 at the LOAEL dose along with the EE group (Figure 7B). However, no significant differences were observed in F3 generation (Figure 7C).

## 3. Discussion

In the current study, we investigated the transgenerational effects of BPA exposure on the physiology, health, and stemness characteristics of testicular germ cells, and testicular morphology in adult mice. With the relatively long period (six weeks) of exposure in adult age, we anticipated that the effects would be prevalent mostly in F0 generation (exposure groups), with the possibility of transgenerational transmission. Hence, all generations (F0–F4) were considered for the experiments. In our initial set of experiments, we did not find any significant difference in the testicular weight (data has not been shown) on BPA exposure. Therefore, the next step included a detailed examination of the testes from the exposed groups (F0) and subsequent generations (F1–F4), to detect any abnormalities in the STs. As shown in Figure 2B–D, a high percentage of abnormal tubules were found at almost all BPA doses in F0–F2, except for the NOAEL group in F1 generation (Figure 2C). Therefore, it was speculated that the high frequencies of abnormal tubules due to BPA exposure could have a crucial effect, although the subsequent generations (F3–F4) showed recovery from the effects of the exposure, and no testicular abnormalities were observed in the subsequent generations.

A thorough examination of the tubular abnormalities showed that some of the lumen areas were abnormally large, resulting in a relatively narrow area for the SE. Our subsequent studies focused on the BPA-induced alterations in the SE and changes in the size of the STs and lumens. As the volume of tubules and lumens differ at various stages of the SE [37], the measurements were taken according to tubular staging. As shown in Appendix A, no significant difference was observed in the area of tubules. However, a significantly large lumen area was observed at stages VII and VIII of the LOAEL-exposed group in F0 and F1. These effects were absent in F2–F4 generations. Moreover, BPA-induced effects on the SE staging were also evaluated for all generations. We observed a high frequency of stage VII together with a low frequency of stage VIII at the NOAEL and LOAEL doses in F0 and at the LOAEL dose in F1 and F2 (Figure 3A–C). Although the reasons for the BPA-induced inconsistencies of frequencies between stages VII and VIII are unclear, it can be postulated that the reduction in the percentage of stage VIII tubules could be because of a delay in the process of spermiation that occurs at stage VIII [24]. This is consistent with the results of a previous study that showed a BPA-induced reduction in sperm counts [24]. However, F3 and F4 generations showed recovery from the abnormalities related to SE staging.

Next, we examined the BPA-induced changes in the proportion of testicular germ cells, abnormalities in testis morphology, and differences in SE stages. Generally, spermatogonia are classified into three categories—type A, intermediate, and type B spermatogonia, which appear during SE stages I–VI [38]. The calculated ratio of spermatogonia to Sertoli cells from each ST showed that a reduction was seen only in the BPA exposure group (F0) (Figure 4B), while the other generations (F1–F4) did not show any changes. Next, we studied the changes in the total number of cells seen during spermatogenesis and examined them using the FACS analysis. As shown in Figure 5, F0 and F1 generations showed differences in the proportions of germ cells regarding 1C-, 2C-, and 4C-type cells. An increase in the total number of spermatids (1C) was seen in F0–F2 at the LOAEL dose, which could be because of the relatively high number of SEs at stage VII. It was also observed that many of the STs that lacked lumen were filled with cell mass, with their appearance reminiscent of round-shaped spermatids (Figure 2A). The frequency of tubular abnormality was also found to be high at the LOAEL dose (Figure 2B–D), which could be the reason for the high count of 1C-type cells at that dose. However, F0 and F1 showed a low count of 2C- and 4C-type cells at the LOAEL dose. Based on these findings, it can be concluded that BPA exposure transgenerationally alters the germ cell proportion and can hamper spermatogenesis.

These results also led us to investigate whether apoptosis occurred among the populations of BPA-exposed germ cells. After BPA is administered subcutaneously to mouse pups from postnatal day (PND) 1 to 21, at PND 22 the pups show an increase in the frequency of apoptotic STs [32]. Similarly, in this study (Figure 6), we observed a high percentage of apoptotic germ cells in the STs and a significantly high number of apoptotic germ cells per ST at the NOAEL and LOAEL doses in F0. In F1 and F2, the effects were limited to only the LOAEL dose and were further diminished in the next generations (F3 and F4). Therefore, BPA exposure can induce abnormal cell death transgenerationally, in addition to causing alterations in the proportions of germ cell populations.

The final objective of this study was to examine the effect of transgenerationally BPA exposure on stemness properties of SSCs. For this, germ cell transplantation was performed in busulfan-treated recipient CD-1 male mice. The germ cells were collected from F2 and F3 generations, since these generations were nearly free from the adverse effects of BPA exposure. The results demonstrated a significant decrease in the donor germ cell-derived colony number at the LOAEL dose in F2, with recovery seen in F3 generation (shown in Figure 7). Therefore, it can be concluded that BPA exposure can reduce the stem cell-like characteristics of SSCs until F2 generation, with no effect on the number of total spermatogonia (Figure 4D). 

To the best of our knowledge, this is the first comprehensive in vivo study on the effects of BPA exposure during the adult period in mice, with the exposed males being examined transgenerationally for testicular morphology, germ cell count, and functional properties of SSCs. The NOAEL dose showed certain adverse effects, especially in the F0 exposure group. BPA induced changes in almost all of the tested parameters in F0–F2 generations at the LOAEL dose, which were subsequently abolished in the next generation (F3). Thus, BPA exposure in the adult period can induce widespread disruption in reproductive health, even at the permissible exposure limits. Since this is a matter of public health and safety, it is crucial to reevaluate the exposure limits of BPA that are deemed to be non-hazardous.

## 4. Materials and Methods

### 4.1. Experimental Animals

Breeding stocks of male and female ICR (CD-1) mice were purchased from Dooyeol Biotech (Seoul, Korea). Recipient CD-1 male mice were used for testicular germ cell transplantation. All animal handling and experimental protocols were approved by the “Animal Care and Use Committee of Chung-Ang University” (IACUC Number: 2016-00009, 26, Feb, 2016) and the “Guide for the Care and Use of Laboratory Animals” published by the National Institutes of Health.

### 4.2. Experimental Design, BPA Exposure, and Breeding to Generate F1–F4 Offspring

After 1 week of adaptation, male mice (F0) were administered BPA (239658, Sigma-Aldrich Chemical Co., St. Louis, MO, USA) by gavage for six weeks at two dose levels of 5 mg/kg bw/day (NOAEL) and 50 mg/kg bw/day (LOAEL). Ethinylestradiol (EE), an orally active estrogen, served as a positive control at a dose of 0.4 µg/kg/day [39]. Both BPA and EE were dissolved in corn oil (Sigma, St, Louis. MO, USA). Control animals were administered corn oil alone [40]. Each of the experimental and control groups consisted of 12–15 male mice. 

In the week following gavage feeding, BPA-exposed male mice were placed with females at a ratio of 1:2 for mating. Females were checked for vaginal plugs to confirm pregnancy. Once pregnancy in the female mice was confirmed by the presence of vaginal plugs, the males were sacrificed for testes collection. The F1 progeny of the pregnant females were allowed to have a three-week lactation period, following which the males were separated and reared for ~95 days before being mated with outbred females. The further week-long mating and sampling procedures followed were similar to that for F0. The process was repeated until F4 generations were obtained (Figure 1B).

### 4.3. Collection of Testes and Determination of Testicular Abnormalities

The testes of F0–F4 male mice were collected, weighted, and dissected vertically into two parts; one each for paraffin sectioning and fluorescence-activated cell sorting (FACS) assay. For paraffin sectioning, the testicular parts were fixed in Bouin’s solution (Sigma, St, Louis. MO, USA) for 6 h at room temperature, followed by washing with 70–100% ethanol gradient at 5 min intervals for dehydration. The tissue was subsequently washed in xylene and embedded in paraffin wax. Tissue sections of 5 µm thickness were cut and placed on glass slides, some of which were stained with hematoxylin and eosin (H&E) for testicular morphology study and examined under a fluorescence microscope (TE2000-U, Nikon, Chiyoda-ku, Tokyo, Japan). The abnormalities in the testis were classified into several categories: The presence of seminiferous tubules (STs) with huge lumen or with no lumen, presence of abnormal cell mass inside lumen, loss of germ cell in the SE, and presence of vacuole in the SE [40]. All the STs present in one section of the testicular tissue were taken into account while obtaining data.

### 4.4. Periodic Acid–Schiff (PAS)–H&E Staining, Staging of SE, and Counting of Spermatogonia

For the staging of SE, 5 µm-thick sections of the testes were stained with PAS and counterstained with H&E [38]. For examination of the sections and measurement of areas of the ST and lumen, a Nikon TE2000-U microscope and NIS-Elements imaging software (Nikon, Chiyoda-ku, Tokyo, Japan) were used.

All types of spermatogonia, namely, type A, type B, and intermediate, were observed visually in the SE at stages I–VI [38]. The total number of spermatogonia was counted, together with the number of Sertoli cells from the same ST, and the ratio of spermatogonia per Sertoli cells was obtained.

### 4.5. Flow Cytometric Analysis

Populations of testicular cells, which are classified as 1C, 2C, and 4C subpopulations based on their DNA content, were measured using FACS. Before FACS, testicular cells were separated from the testis tissue according to Oatley and Brinster [41]. In brief, after removing the tunica albuginea from the testis, the tissue (containing STs) was treated enzymatically with collagenase (1 mg/mL; Gibco, Carlsbad, CA, USA) for 1–2 min at 37 °C with a slight agitation. A solution containing 4:1 ratio of 0.25% trypsin-ethylenediaminetetraacetic acid (Gibco, Carlsbad, CA, USA) and DNase I dissolved in Dulbecco’s phosphate buffered saline (DPBS; 7 mg/mL; Roche, Mannheim, Germany) was prepared and used to treat the STs for 5–6 min at 37 °C with flick mixing. Thereafter, fetal bovine serum (FBS) (10% [*v/v*], Biotechnics Research, Lake Forest, CA) was added to inactivate the enzymatic action, and a cloudy suspension was prepared by soft pipetting. The cell suspension was filtered using a 40 µm pore size nylon mesh (BD Biosciences, San Jose, CA, USA). Cells were fixed with 70% chilled ethanol and stored at 4 °C overnight. The fixed cells were washed two times with chilled DPBS (Gibco, Carlsbad, CA, USA) and treated with DPBS containing 500 µg/mL RNase (Sigma-Aldrich, St, Louis. MO, USA) and 0.1% Triton X-100 (*v/v*, in DPBS) for 15 min at 4 °C in the dark. Finally, the treated cells were stained with propidium iodide (Sigma, St. Louis, MO, USA) and analyzed using a FACS Aria II Flow Cytometer (BD Biosciences, San Jose, CA, USA) with Cell Quest software (BD Biosciences, San Jose, CA, USA), according to Liu et al. [24]. 

### 4.6. Detection of Germ Cell Apoptosis

Paraffin-embedded testis sections, which had not been stained with H&E, were used for the detection of apoptotic germ cells in the SE by the terminal deoxynucleotidyl transferase dUTP nick-end labeling (TUNEL) assay using the in situ cell death detection kit, POD (Roche, Mannheim, Germany), following the manufacturer’s protocol. Since apoptotic cells exhibit green fluorescence, they were visualized under a fluorescence microscope (TE2000-U, Nikon, Chiyoda-ku, Tokyo, Japan). The number and percentage of STs positive for apoptotic cells were measured. 

### 4.7. Germ Cell Transplantation to Evaluate the Activity of SSCs

Transplantation of germ cells into recipient testis is an established method for evaluating the number of functional SSCs in the testis of the donor animal and was performed as described previously [41,42]. Briefly, six-week-old CD-1 male mice were used as the recipient mice. They were prepared for transplantation by the injection of busulfan (35 mg/kg bw; Sigma, St, Louis. MO, USA) into the intraperitoneal cavity (IP) after six weeks used for transplantation. 

Germ cells were then stained using the PKH26 red fluorescent cell linker kit (Sigma, St, Louis. MO, USA) following the manufacturer’s protocol. The proportion of SSCs is 0.03% of the total cells in the testis [43]; therefore, a suspension of 50 × 106 cells/mL was prepared with 10% (*v/v*) FBS and 10% DNase I (7 mg/mL) in a minimum essential medium α (12000-014, powder, Gibco, Carlsbad, CA, USA). Recipient mice were anesthetized with ketamine (75 mg/kg) and medetomidine (0.5 mg/kg) before transplantation. Hair was removed from the lower abdomen, followed by disinfection with iodine and ethanol (70%). A small surgical wound was made, and the testis was pushed out from the abdomen. The suspension of donor germ cells was labeled with 7% (*v/v*) trypan blue and injected into the testes of recipient through efferent ducts as described previously [41]. Each of the testes received ~8–10 µL (~5.0 × 105 cells) cell suspension, which filled approximately 80% of the surface STs.

Analysis of the testes was performed one month after transplantation. The recipient mice were euthanized, and the testes were collected and visualized under a fluorescence microscope (Nikon AZ100, Tokyo, Japan). After the testes were decapsulated and the tubules spread gently, donor cell-derived colonies were visualized separately under the microscope and counted, as described previously [44]. The spherical PKH26-positive colonies (length ≥ 200 µm) were considered to have been produced from one SSC and were counted. Histological examination of the testes using fluorescence microscopy was also performed to confirm the proliferation of spermatogonia at the basement membrane of the recipient STs.

Each group consisted of 8–10 recipient mice, and the donor germ cells were injected into both testes of the recipient. The number of donor-derived colonies was calculated as follows: Colonies/105 transplanted cells = Number of colonies × 105/Number of transplanted cells.

### 4.8. Statistics

One-way analysis of variance was used for analyzing data with the help of SPSS Statistics (version 23.0, IBM, Armonk, NY, USA) and GraphPad Prism (version 5.03; GraphPad Software Inc., La Jolla, CA, USA) software. Significant differences among the mean values were determined using Tukey’s Honestly Significant Difference test. A *p*-value of < 0.05, compared to the control, was considered as statistically significant.

## Figures and Tables

**Figure 1 ijms-21-05408-f001:**
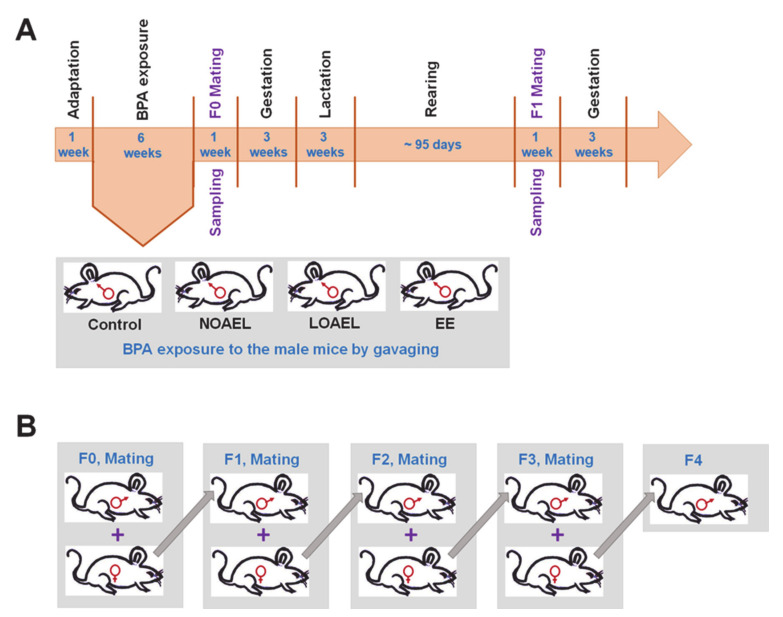
Experimental design. (**A**) After adaptation for one week, CD-1 male mice (F0) were gavaged with different bisphenol-A (BPA) doses [no-observed-adverse-effect-level (NOAEL; 5 mg/kg body weight (bw)/day) and lowest-observed-adverse-effect level (LOAEL; 50 mg/kg bw/day); see detailed dose description in the text], negative control (corn oil), and positive control (ethinylestradiol; EE). (**B**) Mating scheme to produce descendants (F0–F4 generation) where males were allowed to mate with outbred females.

**Figure 2 ijms-21-05408-f002:**
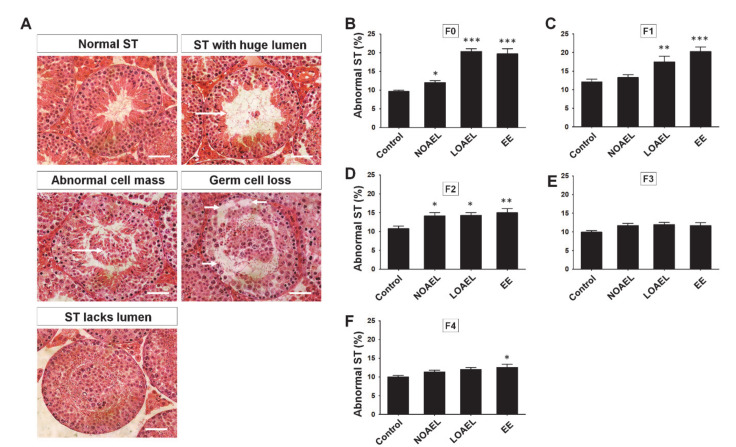
Testicular cross section and abnormal seminiferous tubules (STs) due to bisphenol-A (BPA) exposure. (**A**) Hematoxylin and eosin-stained testicular cross sections in the control, BPA doses [no-observed-adverse-effect-level (NOAEL), lowest-observed-adverse-effect level (LOAEL)], and ethinylestradiol (EE). It was showing normal STs along with abnormal STs, characterized as STs with a large lumen, abnormal cell mass, loss of germ cells, and that lack lumen. Arrows in the images indicate the positions of abnormality. Scale bar = 50 µm. Percentages of abnormal tubules are presented as bar graphs; (**B**) exposed group (F0) (*n* = 14 mice/group), (**C**) F1 generation (*n* = 15 mice/group), (**D**) F2 generation (*n* = 12 mice/group), (**E**) F3 generation (*n* = 12 mice/group), and (**F**) F4 generation (*n* = 19 mice/group). Data were analyzed by one-way analysis of variance (ANOVA) where the asterisk (*) indicates significant differences compared with the control *(** *p < 0.05,* ** *p < 0.01*, and *** *p < 0.001*).

**Figure 3 ijms-21-05408-f003:**
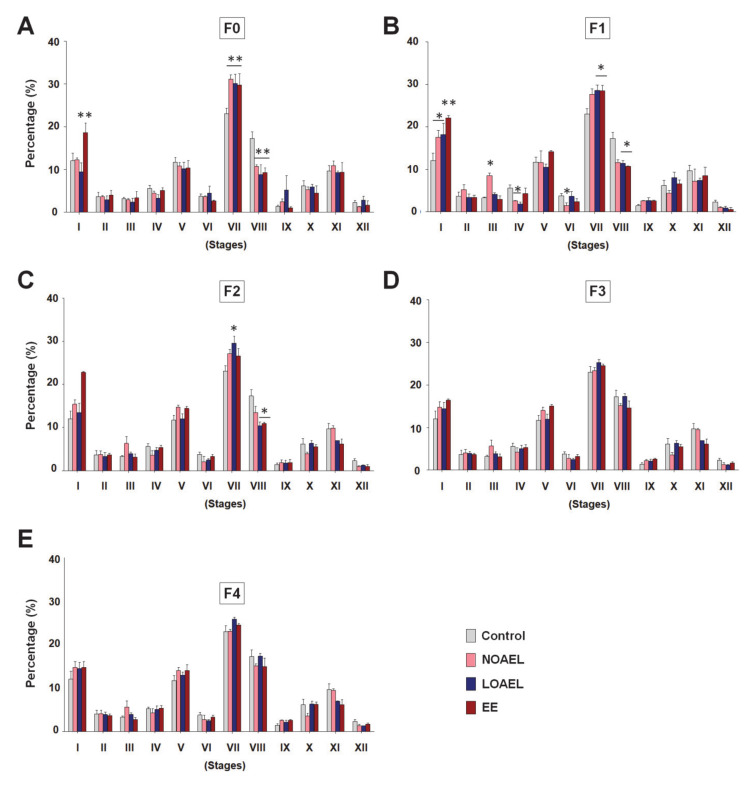
Staging of seminiferous epithelium (SE). Testicular sections were stained with periodic acid–Schiff (PAS)-hematoxylin and eosin, and SE staging was performed to assess the differences due to bisphenol-A exposure. The bar graphs show the staging patterns of (**A**) exposed group (F0) (*n* = 9 mice/group), (**B**) F1 generation (*n* = 9 mice/group), (**C**) F2 generation (*n* = 9 mice/group), (**D**) F3 generation (*n* = 9 mice/group), and (**E**) F4 generation (*n* = 9 mice/group). For each stage, the difference between exposed and control groups was assessed by one-way analysis of variance (ANOVA) (* *p < 0.05* and ** *p < 0.01*).

**Figure 4 ijms-21-05408-f004:**
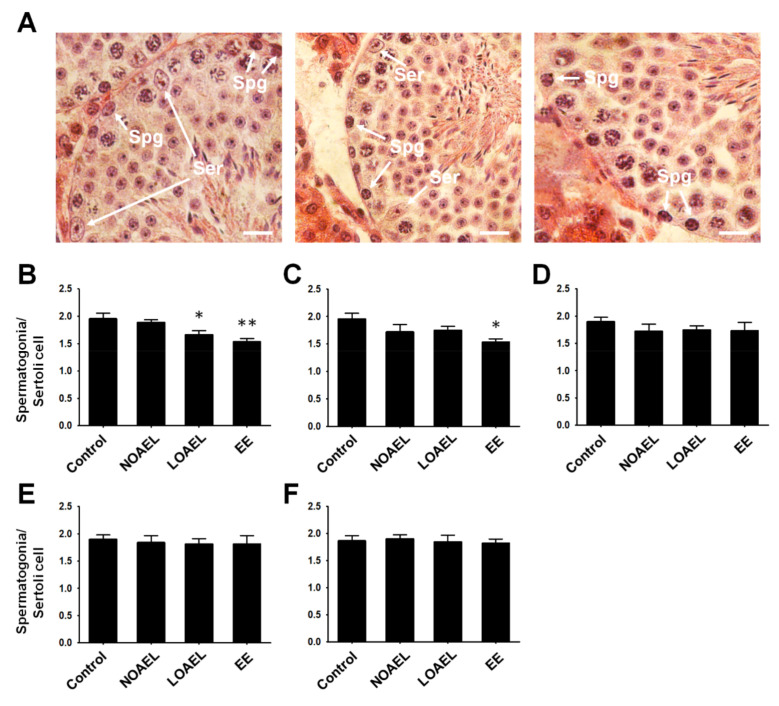
Ratio of spermatogonia and Sertoli cells. Ratios were measured as the total number of spermatogonia per Sertoli cell of the same seminiferous epithelium (SE). (**A**) Testicular cross sections showing SE containing spermatogonia (spg). Scale bar = 10 µm. Ratios of spermatogonia and Sertoli cells (ser) from (**B**) exposed group (F0) (*n* = 6 mice/group), (**C**) F1 generation (*n* = 6 mice/group), (**D**) F2 generation (*n* = 6 mice/group), (**E**) F3 generation (*n* = 6 mice/group), and (**F**) F4 generation (*n* = 6 mice/group). Statistical analysis of the difference between exposed and control groups was performed using one-way analysis of variance (ANOVA) (* *p < 0.05* and ** *p < 0.01*).

**Figure 5 ijms-21-05408-f005:**
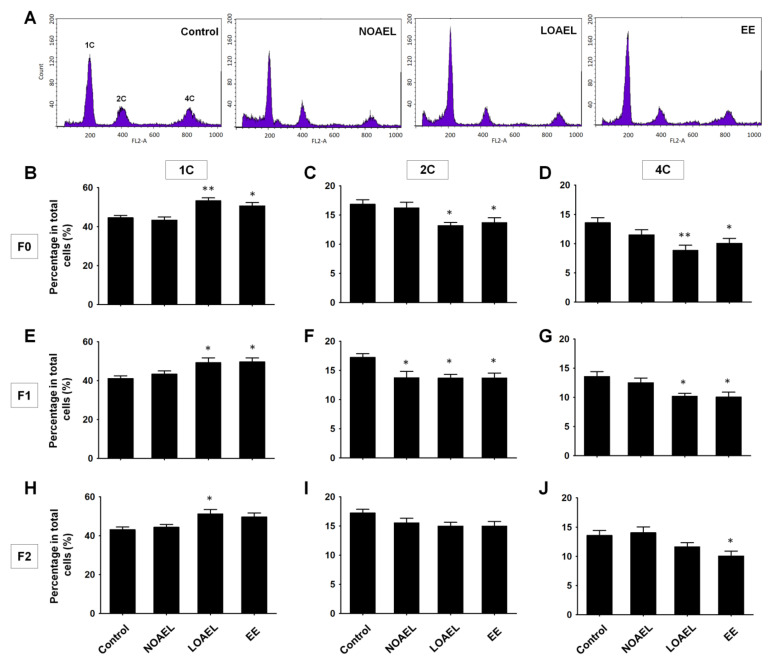
Bisphenol-A (BPA)-induced changes in the proportion of testicular cell populations. (**A**) Representative images of the control, BPA doses [no-observed-adverse-effect-level (NOAEL), lowest-observed-adverse-effect level (LOAEL)], and ethinylestradiol (EE), derived from fluorescence-activated cell sorting (FACS). Letters in the images (1C, 2C, and 4C) indicate the proportion of haploid (1C), diploid (2C), and tetraploid (4C) germ cells in the testicle. In the case of exposed group (F0), bar graphs are showing the percentage of (**B**) haploid, (**C**) diploid, and (**D**) tetraploid germ cells among the total cells. Percentage of germ cells in F1 generation; (**E**) haploid, (**F**) diploid, and (**G**) tetraploid. Percentage of germ cells in F2 generation; (**H**) haploid, (**I**) diploid, and (**J**) tetraploid. Data were analyzed by one-way analysis of variance (ANOVA) where significant differences (*) were calculated as the comparison of exposed group with the control (*n* = 10 mice/group; * *p < 0.05* and ** *p < 0.01*).

**Figure 6 ijms-21-05408-f006:**
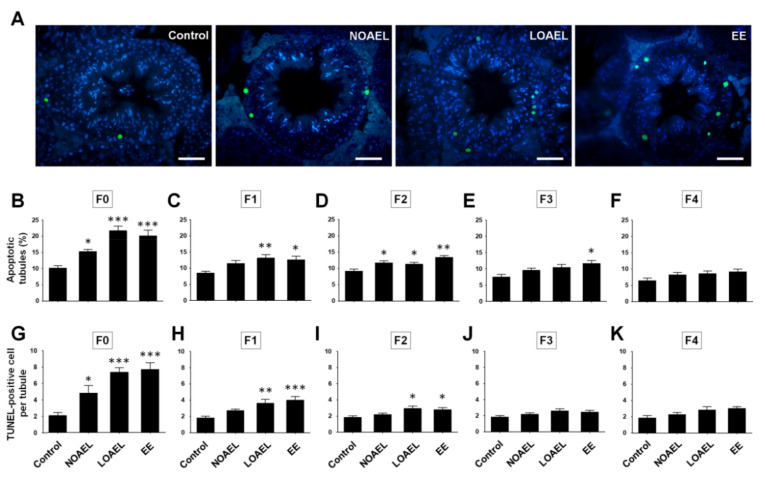
Bisphenol-A-induced apoptosis on testicular germ cells. The terminal dUTP nick-end labeling (TUNEL) assay was utilized for the detection of apoptotic germ cells in the seminiferous epithelium (SE). (**A**) Images represent the position and number of TUNEL-positive germ cells (green) among the other cells (blue; DAPI-stained) of SE from the control, no-observed-adverse-effect-level (NOAEL), lowest-observed-adverse-effect level (LOAEL), and positive control (ethinylestradiol, EE) groups. Scale bars = 50 µm. Bar graph shows the percentages of apoptotic tubules of (**B**) exposed group (F0), (**C**) F1 generation, (**D**) F2 generation, (**E**) F3 generation, and (**F**) F4 generation. Similarly, the graphical presentation of TUNEL-positive germ cells per tubules of (**G**) exposed group (F0), (**H**) F1 generation, (**I**) F2 generation, (**J**) F3 generation, and (**K**) F4 generation are also shown. Data were generated from at least 12 mice/group and analyzed with one-way analysis of variance (ANOVA), where * *p < 0.05*, ** *p < 0.01*, and *** *p < 0.001*, are compared with the control.

**Figure 7 ijms-21-05408-f007:**
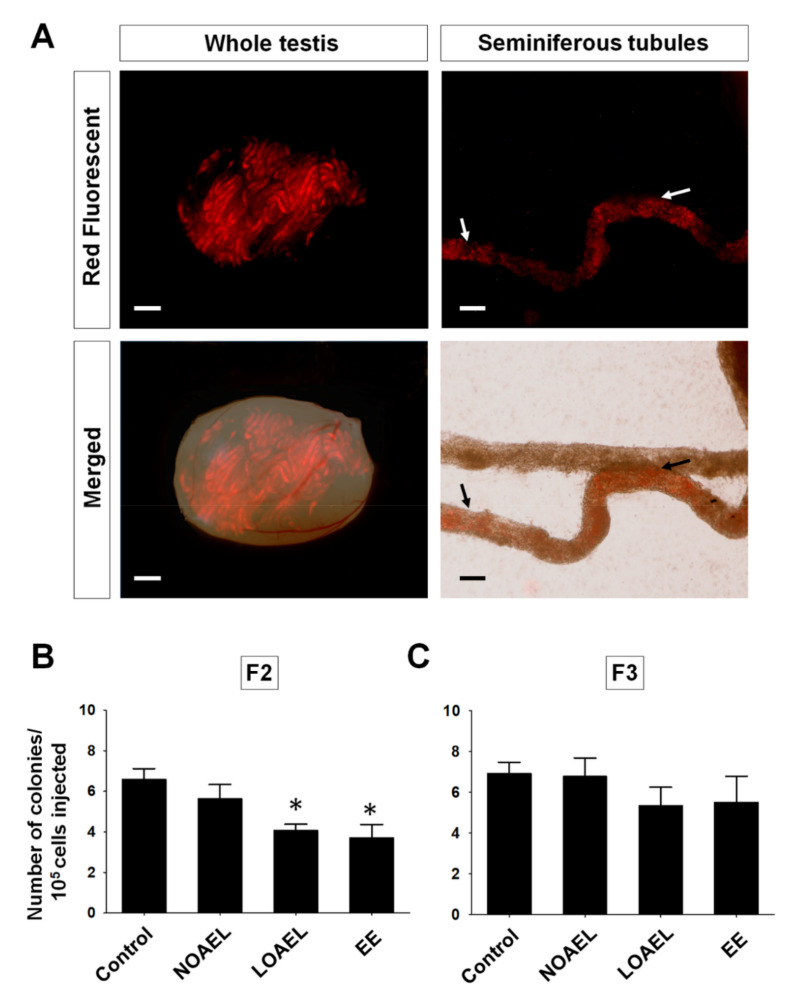
Germ cell transplantation into recipient mice and analysis of recipient testes. Germ cell transplantation was conducted to evaluate the bisphenol-A-induced effect on the stemness properties of spermatogonial stem cells (SSCs). Donor germ cells collected from F2 and F3 generations were stained with PKH26 (red fluorescent). (**A**) Images of the left panel show fluorescent expression of seminiferous tubules (STs) and the merged image of STs inside the recipient testis of transplanted germ cells for F2 generation (scale bar = 2 mm). The right panel shows a fluorescent and merged view of the colonization of donor germ cells inside the recipient ST. White and black arrows are showing the places of colonization. Scale bar = 200 µm. The bar graph represents the number of donor cell-derived colonies per 10^5^ transplanted germ cells for (**B**) F2 and (**C**) F3 generations. For the F2 generation, the total numbers of mice/testes analyzed were 9/15, 9/16, 10/18, and 8/16 for the control, no-observed-adverse-effect-level (NOAEL), lowest-observed-adverse-effect level (LOAEL), and ethinylestradiol (EE), respectively. For the F3 generation, the total numbers of mice/testes analyzed were 8/15, 9/17, 9/16, and 8/15 for the control, NOAEL, LOAEL, and EE, respectively. The asterisk (*) indicates significant differences (* *p < 0.05*) compared with the control.

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
