# Peer review of "Paternal Exposure to Bisphenol-A Transgenerationally Impairs Testis Morphology, Germ Cell Associations, and Stemness Properties of Mouse Spermatogonial Stem Cells"

_ijms, 2020, doi:10.3390/ijms21155408_

Round 1
Reviewer 1 Report
In the manuscript “Paternal Exposure to Bisphenol-A Transgenerationally impairs Testis Morphology, Germ Cell Associations, and Stemness Properties of Mouse Spermatogonial Stem Cells”, Karmakar et al. describe an approach to scrutinize the effects of bispenol A (BPA) on directly imposed individuals and following generations (up to F4) after paternal exposure. Therefore, adult male mice were administered to two different doses of BPA (no-observed-adverse-effect-level (NOAEL) and lowest-observed-adverse-effect level (LOAEL)) for six consecutive weeks. Testes of the sires (F0 generation) and after breeding the male descendants up to the F4 generation were investigated for potential seminiferous epithelium (SE)/seminiferous tubule (ST) degeneration, changes in the germ cell composition and apoptosis as well as spermatogonial stem cell (SSC) stemness properties. For this purpose, methods such as light microscopy in combination with HE-staining, TUNEL, FACS and SSC transplantation were employed.
The analyses revealed that BPA higher dose (LOAEL) effects were transmitted to the next generations up to F2, but were ablated in F3 and F4. In contrast, doses of the BPA NOAEL only showed effects in F0.
This research addresses an important topic, since BPA, that has been proven to affect the germ line due to its hormone-like properties, is used extensively for products of human daily life nowadays. The study has been carefully conducted, nevertheless, some aspects regarding the manuscript still should be improved (summarized below).
Line 27: The abbreviation “LOAEL” has to be already introduced here.
Line 48: “exhibit”
Lines 50-51: The statement about the spermatozoa as the final products of the SSCs should be formulated a little bit more extensive (2 sentences) and more precisely:
- Spermatozoa are produced FROM, but not IN spermatogonial stem cells!
- The balance of self-renewal and differentiation should be directly related to the SSCs.
All Latin terms (in vivo, in vitro) should be written in italics (lines 47, 56, 66, 287).
Lines 53-56: Sentence with 5x “and”. Please rephrase!
Line 64: “of the proteome”
Lines 71-72: “of the size”
Line 78: “outbred females” (Plural)
Lines 89, 169, 195, 222: “Ethinylestradiol” as in line 312.
Line 88 and ff: "Corn oil alone" should not only be declared as "control", but as "negative control" analogue to the designation “positive control”. Please change throughout the manuscript.
Line 93: The abbreviation "STs" has not yet been introduced here (is first introduced in Fig. 2).
Line 94: Better: “For detection of testicular abnormalities,…”
Line 95: Better: “STs were characterized as abnormal in the case of huge lumen size, …”
Line 123: Missing space after the sentence.
The title of the chapter starting in line 138 and the beginning of the chapter suggest that here ratios of Sertoli cells to the DIFFERENT TYPES of spermatogonia were measured. However, text and Figure 4 only refer to spermatogonia in general. If the examination was actually performed in relation to the different types of spermatogonia, these data should be shown. Otherwise, heading and corresponding sentences should be edited. (In addition, for a reliable cell type classification usually different cell types should be stained with cell-specific markers, in this case a Sertoli cell marker and a pan-spermatogonia marker.)
Line 155: “was also studied”
Lines 213/214 (Fig. 7): Which generation is shown in Fig. 7A, F2 or F3? This information should be added under (A) in the figure legend.
From line 219: The formatting of "105" and "106" is incorrect. Please modify throughout the manuscript!
Line 301: “Dooyeol Biotech”
Line 390: “disinfection” instead of “disinfestation”
Line 421: Missing full stop at the end of the author list.
Fig. 2A: Please show these pictures in larger size. Otherwise, it is not possible to identify, for example, the round spermatids, as indicated in line 266 (Discussion).
Fig. 2B-F: The explanation for NOAEL, LOAEL and EE is missing in this figure legend. Please add as in Fig. legend 1.
Fig. legend 3D: n (x mice/group) for F3 is missing.
Table S1: For the purpose of completeness and for a better overview, the values for the generations F0-F2 should also be shown here.
Fig. S1: The formatting for square µm is incorrect → µm2.
Reviewer 2 Report
In the abstract (line 27) the term LOAEL should be clarified.
To respect the instructions for the authors described on the website "research manuscript sections, Figure 1 (experimental design) should be included in material and methods and not at the end of the introduction section.
Also, some grammar considerations should be taken into account.
Among others:
Line 8: Please consider capitalizing the word “institute”.
Line 48: The word “exhinit” is not correct. Please consider changing.
Line 51: It seems that the verb “appears” does not agree with the subject. Please consider changing the verb form.
Line 51: It seems that the article use may be incorrect here. Please consider changing.
Line 75: The noun phrase “association” seems to be missing a determiner before it. Please consider adding the article.
Line 76: The noun phrase “testis” seems to be missing a determiner before it. Please consider adding the article.
Line 78: It seems that “female” may not agree in number with other words in this phrase. Please consider changing
Line 95: It seems that the preposition “with” use may be incorrect here. Please consider changing.
Line 155: The plural verb “were” does not appear to agree with the singular subject. Please consider changing the verb form for the subject-verb agreement.
Line 199: It appears the sentence uses an incorrect for of the verb “showed”. Please consider changing it.
Line 247: It seems that the verb “was” does not agree with the subject. Please consider changing the verb form.
Line 266: It seems that “spermatid” may not agree in number with other words in this phrase. Please consider changing.
Author Response
In the abstract (line 27) the term LOAEL should be clarified.
Response: Changed from “LOAEL” to “lowest-observed-adverse-effect level (LOAEL)”
To respect the instructions for the authors described on the website "research manuscript sections, Figure 1 (experimental design) should be included in material and methods and not at the end of the introduction section.
Response: We thank the reviewer for this comment to improve our manuscript. However, according to the IJMS journal's contribution policy, the Material and Methods section is located behind the manuscript, I think the flow of the paper can be strange if the experiment design figure goes backward. therefore Figure 1 was placed behind the introduction section to help readers understand.
Also, some grammar considerations should be taken into account.
Among others:
Line 8: Please consider capitalizing the word “institute”.
Response: Changed from “institute” to “Institute”.
Line 48: The word “exhinit” is not correct. Please consider changing.
Response: Changed from “exhinit” to “exhibit”.
Line 51: It seems that the verb “appears” does not agree with the subject. Please consider changing the verb form.
Response: Changed from “institute” to “Institute”.
Line 51: It seems that the article use may be incorrect here. Please consider changing.
Response: Spermatozoa are produced from spermatogonial stem cells (SSCs) as the final products of spermatogenesis, which appears to balance of self-renewal and differentiation should be directly related to the SSCs” change completed.
Line 75: The noun phrase “association” seems to be missing a determiner before it. Please consider adding the article.
Response: It means "BPA exposed association of germ cell".
Line 76: The noun phrase “testis” seems to be missing a determiner before it. Please consider adding the article.
Response: Changed from “testis” to “testes”.
Line 78: It seems that “female” may not agree in number with other words in this phrase. Please consider changing
Response: Changed from “female” to “females”.
Line 95: It seems that the preposition “with” use may be incorrect here. Please consider changing.
Response: Changed from “with huge lumen size” to “as abnormal in the case of huge lumen size”.
Line 155: The plural verb “were” does not appear to agree with the singular subject. Please consider changing the verb form for the subject-verb agreement.
Response: Changed from “were” to “was”.
Line 199: It appears the sentence uses an incorrect for of the verb “showed”. Please consider changing it.
Response: Changed from “Showed” to “shown”.
Line 247: It seems that the verb “was” does not agree with the subject. Please consider changing the verb form.
Response: Changed from “was” to “were”.
Line 266: It seems that “spermatid” may not agree in number with other words in this phrase. Please consider changing.
Response: Changed from “spermatid” to “spermatids”.